# Tracking the Evolution of Cutaneous Melanoma by Multiparameter Flow Sorting and Genomic Profiling

**DOI:** 10.3390/ijms26041758

**Published:** 2025-02-19

**Authors:** Luca Roma, Thomas Lorber, Sabrina Rau, Michael T. Barrett, Caner Ercan, Federica Panebianco, Salvatore Piscuoglio, Katharina Glatz, Lukas Bubendorf, Christian Ruiz

**Affiliations:** 1Institute of Medical Genetics and Pathology, University Hospital Basel, University of Basel, CH-4031 Basel, Switzerland; 2Department of Research, Mayo Clinic in Arizona, Scottsdale, AZ 85259, USA; 3IRCCS Humanitas Research Hospital, 20089 Rozzano, Milan, Italy

**Keywords:** FACS, WES, melanoma, bioinformatics, omics data

## Abstract

Intratumoral heterogeneity and clonal evolution are pivotal in the progression and metastasis of melanoma. However, when combined with variable tumor cellularity, intratumoral heterogeneity limits the sensitivity and accuracy of uncovering a cancer’s clonal evolution. In this study, we combined fluorescence-activated cell sorting (FACS) with whole-exome sequencing (WES) to investigate the clonal composition and evolutionary patterns in seven melanoma biopsies obtained from three patients, each having both primary site and metastatic samples. We employed a multiparameter ploidy sorting approach to isolate tumor populations based on DNA ploidy and melanoma biomarkers (SOX10 or S100), enabling us to investigate clonal evolution with high resolution. Our approach increased the mean tumor purity from 70% (range 19–88%) in unsorted material to 91% (range 87–96%) post-sorting. Our findings revealed significant inter- and intratumor heterogeneity, with one patient exhibiting two genomically distinct clonal tumor populations within a single primary site biopsy, each giving rise to different metastases. Our findings highlight the critical role of intratumoral heterogeneity and clonal evolution in melanoma, especially when analyzing tumor trajectories. The unique combination of multiparameter FACS and WES provides a powerful method for identifying clonal populations and reconstructing clonal evolution. This study provides valuable insights into the clonal architecture of melanoma and lays the groundwork for future research with larger patient groups.

## 1. Introduction

The genomic landscape of melanoma features numerous genomic alterations with key driver mutations in genes, such as *BRAF* (45%), *NRAS* (17%), and *KIT* (9%), leading to dysregulated cell growth and survival pathways [1,2]. Detecting *BRAF* mutation is vital in advanced melanoma patients for the effective use of BRAF–MEK inhibitors. Despite their success, long-term survival rates remain relatively low [3,4]: only 22% and 19% of the patients treated with first-line BRAF–MEK inhibitors reached 3-year survival and 4-year survival, respectively [3]. Immune checkpoint inhibitors (ICIs) represent a breakthrough in the treatment of melanoma, with long-term responses and even cure in patients with metastatic disease. Thus, ICIs have become the standard of care in BRAF wild-type melanoma and in BRAF mutated melanoma after resistance to BRAF inhibitors. Additional mutated driver genes in melanoma other than *BRAF*, such as *NF1* and *NRAS*, might also impact the treatment choice [5]. As genomic analysis of known cancer genes has become standard practice, typically employing Next-Generation Sequencing (NGS) with small gene or hotspot panels, sufficient tumor material and a tumor cell fraction of 15–20% are needed for reliable mutation detection. However, these routine assays fall short in capturing the clonal composition and intratumoral heterogeneity (ITH) of the tumor. Large-scale research studies performing deep exome or genome sequencing of melanoma biopsies have tried to fill this gap and determine the genomic landscape as well as the evolutionary patterns [6,7,8]. While they have provided fundamental insights into the complex mutational landscape, these studies have only focused on the analysis of single tumor biopsies. Subclonal architecture in single biopsies can nowadays be inferred by applying computational approaches to data from bulk DNA sequencing [9]. However, to infer the evolutionary relationship between different sites of a tumor, multiple biopsies from the same patient are required. The few relevant melanoma studies to date were mainly performed with material from autopsies [10,11,12]. Recently, Spain and colleagues conducted the largest intrapatient analysis of melanoma so far by profiling 573 samples from 14 late-stage melanoma patients who had died despite ICI treatment [13,14]. Interestingly, their study suggested that metastatic cells can start seeding and evolving clonally during the early stage of disease [14]. Although their findings are based on ICI-treated patients, they further emphasize the importance of intratumoral heterogeneity, both between different body sites and within single biopsies, in understanding clinically relevant evolutionary patterns.

While these studies were fundamental for understanding the clonal evolutionary landscape in melanoma, they required resource-intensive techniques, extensive bioinformatic analyses, and substantial tumor tissue typically available only from autopsies. Small biopsy specimens in routine diagnostics do not permit multi-region genomic analysis or macrodissection of tumor regions of interest, but necessitate a more efficient approach to analyze the remaining diagnostic material, including those with low tumor cell content.

We and others have previously shown that DNA content-based flow sorting can identify and sort distinct tumor populations from small biopsies before genomic profiling [15,16]. This approach allowed for the detection of mutations with low variant allele frequency (VAF) otherwise hidden by normal cell contamination and successfully identified specific (private) mutations within distinct tumor populations [15,16]. Since up to 50% of the tumors can be diploid, additional flow-sorting markers are required to differentiate between diploid normal and diploid tumor cells [15,16]. In routine melanoma diagnostics, SOX10 and S100 are commonly used biomarkers for melanoma via immunohistochemistry [17,18]. Here, we developed a refined ploidy-based flow-sorting method that enriches SOX10- or S100-labelled tumor nuclei before genomic profiling. We applied this approach to seven biopsies from three patients with cutaneous melanoma collected for routine diagnostics. This multiparameter ploidy sorting approach followed by whole exome sequencing (WES) on the eight resulting tumor populations allowed us to reconstruct the evolutionary trajectories of these tumors at high resolution across time and different sites.

## 2. Results

### 2.1. Patients

We performed a comprehensive search of the biobank at the Institute of Pathology Basel to identify cutaneous melanoma cases with multiple biopsies from the same patient with at least one biopsy per patient from a metastatic site and the primary site. In order to determine the clonal composition of the biopsies with the highest possible resolution using a multiparameter ploidy sorting approach (see next section), we focused on fresh-frozen bioptic tumor material with a diameter of at least 5 mm. For this study, we selected three patients whose biopsies fulfilled the aforementioned criteria. The average age of the patients was 72.3 (range 63–89), and all patients were female and had been diagnosed with Stage IV melanoma. The composition of the cohort, including its pathological annotation, is summarized in Table 1. Briefly, Patient 1 was diagnosed with malignant melanoma at the age of 63 with a primary tumor on the left thigh. Lymph node metastasis and subcutaneous recurrence of the primary tumor were detected in the same year, and a skin metastasis the year after. Over the subsequent two decades, the patient experienced recurrences at the primary site as well as several distant metastases. For Patients 2 and 3, biopsy material from the primary site and the metastasis, collected in the same year as the primary tumor, were included in this study. No additional information regarding metastases or recurrences in subsequent years was reported in the clinical records of these two patients. Of note, at the time the specified biopsies were taken from Patients 1 and 2, routine DNA flow cytometry analyses were performed on native material in at least one biopsy to determine the so-called DNA index. In our study, historical DNA index information is available for the metastasis collected from Patient 1 (DNA index 1.4, i.e., ploidy of 2.8), and for both biopsies from Patient 2 (each with a DNA index of 0.9, i.e., ploidy of 1.8), as shown in Table 1.

### 2.2. Multiparameter Ploidy Sorting for Detection and Isolation of Tumor Populations

To isolate and genomically profile clonal tumor populations, we applied a multiparameter ploidy sorting approach followed by WES of all fresh-frozen tumors (Figure 1A, Appendix A). This procedure included isolating nuclei from the tumors and flow sorting them by ploidy (using DAPI staining) and markers of tumor origin (SOX10 or S100). This was done to separate tumor populations based on DNA ploidy and to increase tumor purity, respectively. We validated the effectiveness of this approach in distinguishing between SOX10-positive and SOX10-negative nuclei, as well as identifying differences in ploidy, using cell lines (Appendix A).

We applied the aforementioned multiparameter ploidy sorting approach to the seven fresh-frozen biopsies listed in Table 1. Two out of the three primary site tumors were monogenomic, meaning that only tumor cells of a single ploidy were present. The primary site tumor biopsy of Patient 1, however, was polygenomic, as evidenced by the presence of two distinct tumor populations with different ploidies (2.8 and 4.0). Across all biopsies, the ploidies ranged from 1.9 to 4.0. Importantly, the ploidy calculation from the multiparameter ploidy sorting approach was consistent with the ploidy (DNA index) measured at the time of diagnosis for the three biopsies that had this information available (see Table 1 and Table 2). Of note, our multiparameter ploidy sorting approach increased the mean tumor purity from 70% (range 19–88%) of unsorted material to 91% (range 87–96%) after sorting (Figure 1B, Table 2).

### 2.3. Genomic Profiling of Sorted Tumor Populations in Three Melanoma Patients

Four sorted tumor populations from the primary sites, and four sorted populations from the metastases of three patients were subjected to WES. A matched germline control was used for each patient. Among the three patients, Patient 1 had two tumor populations in a primary site biopsy (1PST1 and 1PST2), and unique single populations in one skin (1MT1) and one lymph node (1MT2) metastasis (Figure 1C). The number of synonymous and non-synonymous somatic mutations ranged from 363 in the primary site tumor population of Patient 3 (3PST) to 2173 in one of the two tumor populations of the primary site tumor biopsy from Patient 1 (1PST1) (see Appendix A). For Patients 1 and 2, the majority of the mutations were shared by all tumor populations (Patient 1: 63.3%, 1513/2427; Patient 2: 92.8%, 995/1072). In contrast, the metastatic tumor population of Patient 3 (3MT) exhibited a high proportion of private mutations (69.8%, 813/1164). The average tumor mutational burden (TMB) across tumor populations was 15.62 mutations/Mb (range 3.9–23.3). The highest TMB values were observed in Patient 1 (19.8–23.3 mutations/Mb), followed by the biopsies of Patient 2 (11 and 11.7 mutations/Mb). Patient 3 showed significantly higher TMB values in the metastatic tumor population than in the primary site tumor population (13.1 vs. 3.9 mutations/Mb).

A specific analysis of known cancer driver genes revealed *NRAS* as the most frequently mutated gene (6/8 tumor populations), followed by *KMT2D* (5/8 tumor populations) and *COL5A1* (5/8 tumor populations). All of these mutations were detected in only two patients (P1 and P2). Most driver mutations (71.4%, 10/14) were clonal in primary site tumors and the metastases (Figure 2A). Patient 1 harbored truncal mutations in *KIT*, *MECOM*, *NRAS, KMT2D*, and *TP53* in all four of their profiled tumor populations. Patient 3 had truncal mutations in *BRAF* (V600E) and *PPP6C*. *NF1* mutations were only detected in the biopsies of Patient 2. All of the patients presented a high number of truncal mutations, except for two driver mutations in Patient 1. The *COL5A1* mutation was absent in the metastatic tumor population (1MT2), while the *CTNNB1* mutation was present exclusively in the metastatic tumor population (Figure 2A).

Phylogenetic analysis revealed a significant number of shared mutations among the tumor populations of Patients 1 and 2 originating from a common ancestor. Interestingly, this analysis indicates that in Patient 1, there was an early divergence of the 1MT2 population (lymph node metastasis). In contrast, the 1MT1 population (skin metastasis), diagnosed one year after the lymph node metastasis, is more closely related to 1PST2 (the primary site tumor population with a similar ploidy), suggesting a later stage of divergence. The two tumor populations in Patient 2, consisting of the primary site tumor (2PST) and the lymph node metastasis (2MT), were detected in the same year, shared a high number of mutations, and were of the same ploidy. In contrast, the metastatic tumor population of Patient 3 (3MT) exhibited a large number of private mutations (Figure 2B, right phylogenetic tree).

Additionally, mutational signature analysis indicated that most tumor populations were dominated by UV light signatures (SBS7a and SBS7b). The exception was the breast metastasis from Patient 3 (3MT) which showed a high percentage of signatures related to platinum chemotherapy (SBS31 and SBS32) and alkylating agents (SBS11), likely reflecting the effects of the neoadjuvant chemotherapy treatment before the metastasis emerged. These findings imply that the high number of private mutations in the metastasis in Patient 3 may have been caused and selected for by neoadjuvant chemotherapy (Figure 2C).

Analysis of CNVs in all tumor populations revealed a high level of concordance between the tumor populations within the same patient (Figure 2D). Consistent with the mutational profiling results, the CNV profile in Patient 1 showed that the metastatic tumor population (1MT1) is more closely related to the primary site tumor populations (1PST1 and 1PST2) than 1MT2 (lymph node metastasis). In Patient 3, three genomic loci—9p21 (*CDKN2A* and *MTAP*), 11q13 (*OVOL1*), and 11p15 (*HRAS*)—were homozygously deleted in the primary site tumor population (3PST) but not in the metastatic tumor population (3MT).

### 2.4. Analysis of Sorted Tumor Populations Reveals Distinct Clonal Origins of Metastases

In order to study the clonal evolution within and between the biopsies, we calculated the cancer cell fractions (CCFs) for synonymous and non-synonymous single nucleotide variants (SNVs) using ABSOLUTE [19]. All SNVs were further sub-classified in different clusters according to the similarity of the CCF using PhylogicNDT v.1.0. Clonal (CCF ≥ 0.9) and subclonal mutations (CCF < 0.9) were used to generate a clonal evolution tree (Figure 3A and Appendix A). For Patient 1, we had access to one primary site tumor biopsy and two metastases. Our multiparameter ploidy sorting approach revealed two distinct tumor populations (1PST1 and 1PST2) within the primary site biopsy, providing a unique opportunity to study clonal evolution of different tumor populations within the same biopsy and across metastatic sites in the same patient. In this patient, clonality analysis indicated tumor evolution involving the spread of clones from 1PST1 and 1PST2 to different metastatic sites. We observed that cluster 2 (light blue) with 298 clonal mutations was present in both primary site tumor populations (1PST1 and 1PST2) and the skin metastasis (1MT1), but was absent in the lymph node metastasis (1MT2). In contrast, cluster 4 (black) with 199 clonal mutations was present in only one of the two primary site tumor populations (1PST1) and in the lymph node metastasis (1MT2), but absent in both the other primary site tumor population (1PST2) and in the skin metastasis (1MT1). The same pattern was observed for clusters 5 and 6, with 34 and 227 subclonal mutations, respectively (Figure 3A). These findings suggest that each of the metastases originated from a different tumor population of the primary site: the lymph node metastasis (1MT2) from the 1PST1 population, and the skin metastasis (1MT1) from the 1PST2 primary site tumor population (Figure 3A). Similarly, Pearson correlation analysis of the CNVs yielded comparable results (Figure 3B): 1MT1 had a higher correlation (*R* = 0.99) with 1PST2, while 1MT2 exhibited a higher correlation (*R* = 0.66) with 1PST1 (Figure 3B). In Patient 2, the CCF cluster analysis identified a large cluster (Cluster 1) with 832 clonal mutations (Appendix A), as well as a metastasis-specific cluster (Cluster 6) with only 14 subclonal mutations. In contrast, Patient 3 exhibited a common cluster (Cluster 1) with 330 clonal mutations. Notably, one cluster (Cluster 4) with 10 subclonal mutations was completely absent in the metastasis, while another cluster (Cluster 2), with nine mutations, was clonal (CCF = 1) in the metastatic tumor population.

## 3. Discussion

The advent of NGS technologies has significantly advanced our understanding of intratumoral heterogeneity in cancer. However, most studies with these technologies typically analyze bulk tissue samples, i.e., a mix of tumor and stromal components. This complexity necessitates extensive bioinformatic analyses to deconvolute and interpret the clonal composition of those tumors [20,21].

In this study, we employed a novel customized approach that combines FACS with WES to investigate the clonal composition and evolutionary trajectory of melanoma in three patients, each with biopsies from primary site tumors and metastases. Our multiparameter ploidy sorting approach enabled us to separate melanoma tumor populations based on DNA ploidy (DAPI) and specific melanoma biomarkers (SOX10 and S100) [17,22]. This allowed for a high-resolution analysis of intratumoral heterogeneity and clonal evolution. Our findings uncovered unique evolutionary patterns in each patient. In Patient 1, we identified two genomically distinct tumor populations within the primary site biopsy, each giving rise to a different metastatic clone. This finding demonstrated the capability of our approach to decipher clonal origins and the mechanisms that drive metastasis.

While FACS and NGS are well-established techniques, their combined application in our study represents a unique research approach. Although multiparameter sorting based on ploidy and tumor markers has been applied to other cancer types, this study marks the first instance of its use in melanoma [23]. Historically, DNA ploidy analysis was used in cancer diagnostics, particularly in solid tumors, but its application in melanoma remains rare [24]. By combining this approach with genomics, we effectively addressed the challenge of tumor purity in small biopsies, thereby producing high-quality sequencing data. For instance, although Patient 1 lacked a normal FFPE or fresh-frozen biopsy as a germline control, isolating diploid, SOX10-negative cells from the bulk tumor tissue using FACS allowed us to recover the non-tumoral cell population from the tumor sample. This approach improved the purity of both the tumor and non-tumoral populations, thereby allowing for more effective calling of somatic alterations and improving the overall accuracy of the analysis. We confirmed the absence of tumor cell contamination in those populations by using specific bioinformatic software (DeTiN) [25] (Appendix A). Other technologies, such as single-cell RNA (scRNAseq) sequencing, have been used to determine the intratumoral heterogeneity of melanoma [26]. However, the high costs and technical challenges, such as artefacts introduced by the need to amplify the genome from very low amounts of DNA from a single cell, have restricted the use of single-cell sequencing mainly to scRNA-seq for transcriptional expression profiling rather than to single cell DNA sequencing (scDNA-seq) for mutational profiling. Despite these limitations, our approach may be combined with future advancements in scDNA-seq technologies to reduce costs by focusing sequencing efforts on cells of interest only, such as SOX10-positive melanoma cells.

Our study adds to the limited body of literature on melanoma evolution using paired tumor samples from the same patient [10,11,12]. Earlier studies, such as that by Morita et al. in 1998, relied on less comprehensive methods, including loss of heterozygosity (LOH) analysis of a few selected chromosome regions and *CDKN2A* Sanger sequencing, and primarily focused on clonal relationships between primary tumors and metastases [27]. Their data suggest that heterogeneous tumor cell populations might exist from the early stages of tumorigenesis and evolve independently. This implies that the metastatic progression of melanoma may not follow a linear progression model, although they lacked concrete evidence to support this theory. Our study provides further evidence for this new progression model. Unlike previous studies, we were able to separate distinct tumor populations with different ploidies coexisting within a single primary biopsy. In a recent study, König et al. demonstrated that two separate tumor populations, each exhibiting different morphological phenotypes, of a single melanoma case had varying responses to treatment with adaptive cell therapy involving tumor-infiltrating lymphocytes [28]. Here, we successfully separated two populations from the same primary tumor by using FACS without the need for macro- or microdissection, and showed that these two populations gave rise to different metastases.

Recent large-scale genomic studies in melanoma have primarily focused on mutation profiles predictive of response to treatments such as BRAF, NRAS, and CKIT inhibitors [29]. Our results show a high tumor mutational burden (TMB) across all patients, consistent with the literature on cutaneous melanoma [30]. Importantly, our findings also suggest that TMB can vary significantly between primary and metastatic sites, as observed in Patient 3, where the metastasis exhibited a significantly higher TMB than the primary tumor (13.1 vs. 3.9 mutations/Mb). Phylogenetic analysis of our cohort revealed shared and private mutations across tumor populations, with most mutations being truncal and shared across both primary and metastatic sites in Patients 1 and 2 [12,31]. In contrast, Patient 3 presented a high proportion of private mutations in the metastasis, likely driven by neoadjuvant chemotherapy, as indicated by the mutational signatures associated with platinum and alkylating agents (SBS31, SBS32, and SBS11). This further suggests that chemotherapy treatment can significantly alter the genomic landscape of the tumor, including subclonal mutations [32].

We also observed a new clonal evolution pattern not accounted for by traditional linear or parallel progression models, which describe the late dissemination of the most advanced clone or the early dissemination and separate clonal evolution of the metastasis and primary tumor, respectively. In Patient 1, clonality analysis revealed a tumor evolution involving the coexistence of two distinct tumor populations (1PST1 and 1PST2) in the primary tumor and the spread of clones from those two distinct tumor populations to different metastatic sites. Mutational and copy number variation (CNV) analyses further supported that the majority of alterations were shared across primary site and metastatic tumor populations, suggesting co-evolution of different clones within the primary tumor and late dissemination of those two different clones that formed the metastases. This highlights the importance of accounting for intratumoral heterogeneity when studying clonal evolution in melanoma. 

Our findings may have important implications for melanoma diagnostics. For instance, the distinct clonal origins of metastases in Patient 1 indicate that the mutational profiling of both primary and metastatic sites can offer a more complete understanding of tumor heterogeneity. This approach is particularly relevant for studying the trajectories and evolution of highly heterogeneous tumors, as the detailed insights achieved through high-resolution methods would likely be missed with NGS analysis of unsorted bulk tumor tissue. By combining FACS and NGS to uncover clonal diversity, researchers may gain a better understanding of the trajectories and clonality dynamics underlying highly heterogeneous tumors with multiple metastases. Furthermore, our study emphasizes the potential role of homozygous deletions in genes such as *CDKN2A* and *MTAP* at chromosome 9p21.3, which were observed in the primary tumor of Patient 3. These genomic alterations are known to contribute to the progression of melanoma and other cancers and may have diagnostic or therapeutic relevance [33,34,35].

This study is, to the best of our knowledge, the first to employ flow sorting of nuclei derived from melanoma tissues based on both ploidy and SOX10- or S100-positivity, followed by WES of the sorted tumor populations to investigate the genomic intratumoral heterogeneity and the evolutionary trajectory of cutaneous melanomas. The primary limitation of our study is the small sample size of three patients, which restricts the generalization of our findings. Additionally, potential technical limitations in FACS sorting and inherent biases associated with the multiparameter ploidy approach cannot be excluded, as the sorting process depends on the specificity of melanoma-specific markers (SOX10 or S100) and DAPI to define the populations of interest, which were subsequently analyzed by WES. However, the WES data of sorted populations provided evidence that our approach can identify genomically distinct tumor populations even within a single biopsy and improves tumor purity. Despite these limitations, we gained novel insights into the clonal dynamics of melanoma progression, laying the foundation for future studies with larger sample sizes and different melanoma subtypes. Ultimately, our approach offers a powerful tool for investigating the dynamics of melanoma evolution and intratumoral heterogeneity, with potential applications in research settings.

## 4. Materials and Methods

### 4.1. Clinical Cohort

This study was approved by the Ethics Committee of Northwestern and Central Switzerland (EKBB, No EK47/13). Patient characteristics are given in Table 1. All samples used in this study were collected for clinical (diagnostic) purposes from patients presenting at the University Hospital of Basel with cutaneous melanoma. Samples were collected in liquid nitrogen and stored at −80 °C. Surplus tumor tissue is routinely collected in liquid nitrogen at our institution if enough tissue from the same specimen is available for formalin fixation and diagnostics. Samples complying with the inclusion criteria—(i) histologically diagnosed cutaneous melanoma, (ii) multiple biopsies and/or time points available, (iii) sufficient quality and amount of material—were carefully selected and reviewed by two experienced board-certified pathologists (KG and LB). A total of seven fresh-frozen melanoma tumor biopsies from three patients, and their matched non-tumor biopsies (one fresh-frozen and two formalin-fixed) were collected from the archives of the Institute of Pathology and Medical Genetics, University Hospital of Basel (Table 1).

### 4.2. Nuclei Isolation and Multiparameter Flow Sorting

Nuclei were isolated from fresh-frozen tumor samples according to published protocols [15,16,22,36,37,38]. Bulk DNA from formalin-fixed, paraffin-embedded (FFPE) non-tumor samples were utilized as germline controls to identify somatic alterations through WES. Due to the lack of a non-tumoral sample for Patient 1, we utilized the sorted diploid, melanoma marker SOX10-negative cell populations from the primary site biopsy as a germline control for WES. Absence of tumor contamination was demonstrated with the DeTiN software v.1.7.5.9, as described below. Extracted nuclei were stained with DAPI to allow DNA ploidy analysis via FACS. The melanoma markers SOX10 or S100 were used—in addition to DAPI—as a second parameter to discriminate nuclei derived from tumor cells (SOX10- or S100-positive) from nuclei derived from non-tumor cells (SOX10- or S100-negative). SOX10 and S100 expression were verified by immunohistochemistry (IHC) on corresponding FFPE tissue sections using a SOX10 antibody (monoclonal mouse IgG1, Clone#20B7, R&D Systems, Minneapolis, MN, USA) and an S100 antibody (monoclonal mouse IgG2a, clone 4C4.9, Abnova, Taiwan, China), respectively. Isotype controls were used to assess and correct for non-specific binding and background fluorescence. All tumors showed a nuclear expression of SOX10, except for the breast metastasis of Patient 3. The non-tumor FACS-sorted population that was used as a germline control for WES for Patient 1 was SOX10-negative by FACS.

### 4.3. Tumor Purity and Ploidy Assessments

The ploidy (expressed as *N*) of each tumor population was determined via FACS by calculating the ratio of the geometric means of the DAPI signals in the SOX10- or S100-positive tumor populations relative to their corresponding SOX10- or S100-negative, diploid populations, which were considered non-tumor. Tumor purity was evaluated both pre- and post-sorting. For the unsorted material, we utilized the relative percentages of FACS events within the analysis gates of the SOX10- or S100-positive tumor populations (including tumor nuclei in the G2/M phase) compared to the events within the analysis gates of the SOX10- or S100-negative diploid, normal populations. After sorting, tumor purity was assessed computationally using FACETS v.0.5.14 [39].

### 4.4. DNA Extraction and Quantification

FACS-sorted nuclei were subjected to DNA extraction, quantification, and whole genome amplification, as described previously [15]. DNA extraction of the sorted populations and the FFPE biopsies were performed with the Maxwell instrument according to the standard protocols used in diagnostic routine workflow (Promega, Madison, WI, USA). Quantification of the DNA was performed with the Qubit Fluorometer assay according to the manufacturer’s protocol (Life Technologies, Carlsbad, CA, USA).

### 4.5. Library Preparation and Whole Exome Sequencing

Library preparations were performed with the SureSelect Human All Exon V6 Kit (Agilent, Santa Clara, CA, USA) for whole exome capturing according to the manufacturer’s guidelines. Paired-end 100 bp reads were generated on an Illumina NovaSeq 6000. Library preparation and sequencing were conducted by CeGaT (Tübingen, Germany).

### 4.6. Reads Alignment and Variant Annotation

The raw FASTQ data processing workflow was adapted from Roma et al. [40]. Briefly, after the sequencing, reads were aligned against the reference human genome GRCh38 using the Burrows–Wheeler Aligner (BWA, v0.7.12) [41]. The DeTiN software was used to estimate the contamination of the tumors in matched normal samples and in the diploid, SOX10-negative cell population of the primary site biopsy of Patient 1 that was used as a germline control due to the lack of a non-tumoral sample for this patient [25]. SNVs and indels were called using MuTect2 (GATK 4.1.4.1) [42]. SNVs and indels with a VAF < 1% or that were covered by fewer than three reads were discarded if the SNVs were found in two or more sorted populations from the same patient, and a cut-off of two reads was applied [16,40,43]. We further excluded variants identified in at least two of a panel of 210 non-tumor samples, including the non-tumor samples in the current study. Variant annotation was performed by the SnpEff software v.4.1 [44]. The heatmap of non-synonymous mutations was generated using the R package MAFTOOLS v.2.0.16 [45] by selecting the melanoma cancer driver genes from the Bailey et al. database [46]. A detailed listing of all called somatic variants in the distinct tumor populations, including allelic frequency, genomic loci, amino acid change, mutation type, and potential affiliation to a known cancer gene set, is included in the Appendix A.

### 4.7. Copy Number Alterations and Clonality Analysis

Allele-specific copy number variants (CNVs) were identified using FACETS v.0.5.14 [39], and the log ratio relative to ploidy was used to call deletions, loss, gains, and amplifications [40]. The similarity of the CNV between the different samples was calculated by using a Pearson’s correlation implemented in the corrplot R package v.0.95 and drawn using the ggplot R package v. 3.5.1 [47,48]. The cancer cell fraction (CCF) of each mutation was inferred using ABSOLUTE v1.0.6 [19]. Solutions from ABSOLUTE were manually curated to ensure the solution matched the ploidy estimate generated by FACETS [39]. A mutation was classified as clonal if its probability of being clonal was >50% or if the lower bound of the 95% confidence interval of its CCF was >90%. Mutations were considered as subclonal if they did not meet the mentioned criteria [16]. CCF histograms generated by ABSOLUTE were used as the input to PhylogicNDT [49] to find clusters of mutations, infer subclonal populations of cells and their phylogenetic relationships, and determine the order of occurrence of clonal driver events. PhylogicNDT was run using the parameters “Cluster -rb -ni 1000” to cluster and build the phylogenetic tree with 1000 iterations. Data were drawn using ggplot2 in R version 3.6.1 [47]. Clusters with less than 5 mutations were discarded.

### 4.8. Phylogenetic Analysis and Mutational Signatures

A maximum parsimony tree was built for each case using binary presence/absence matrices based on the repertoire of non-synonymous and synonymous somatic mutations, in the biopsies of the tumors, as described by Murugaesu et al. [50]. A mutation that was found in both or more sorted populations of a biopsy was considered as ‘trunk’. Mutations that were detected in only one sorted population of the tumor were classified as ‘branch’ or ‘private’. Mutational signatures were calculated using Mutational Patterns v.3.16.0 [51] by selecting mutational signatures based on the set of 60 mutational signatures from the COSMIC database (“signatures.exome.cosmic.v3.may2019”).

## Figures and Tables

**Figure 1 ijms-26-01758-f001:**
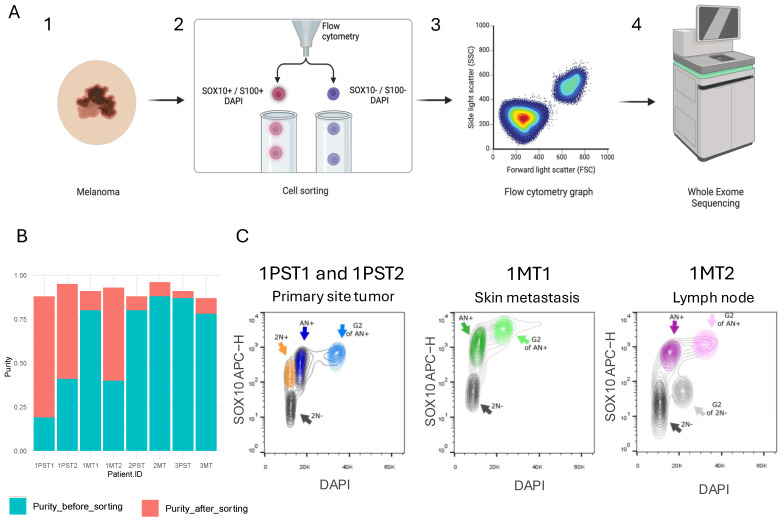
Genomic profiling of flow-sorted tumor populations. (**A**) Schematic overview of the different required steps. 1. Representative melanoma biopsy. 2. Tumor cells were sorted according to Sox10 and S100 positivity/negativity and ploidy (DAPI). 3. Tumor cell populations were collected. 4. Subsequent mutational profiling of tumor populations by WES. Figure created with BioRender.com. (**B**) Bar plot showing tumor purity for each sample before (as determined by FACS) and after FACS sorting (inferred from WES data). (**C**) Figure showing the two distinct tumor populations of Patient 1: 1PST1 (2N+, orange) and 1PST2 (AN+, blue) in the primary site biopsy of Patient 1 and the tumor populations 1MT1 (AN+, green) and 1MT2 (AN+, purple) in the skin and lymph node metastases, respectively. PST = primary site tumor, MT = metastasis tumor. Sorting of the biopsies was performed according to anti-SOX10 positivity (y-axis) and DAPI content (x-axis).

**Figure 2 ijms-26-01758-f002:**
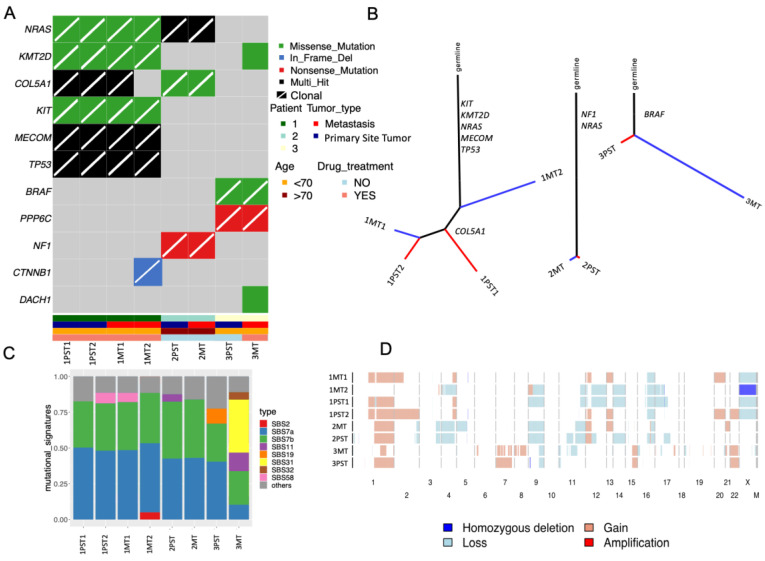
(**A**) Non-synonymous mutation plot showing the presence and distribution of genetic alterations in the eight sorted tumor populations. The genes were selected from the Bailey et al. dataset that represents the cancer driver genes in skin melanomas. The diagonal bar indicates that the mutation is clonal. (**B**) Phylogenetic trees display the number of synonymous and non-synonymous mutations of each patient (Patients 1–3 from left to right). The size of the trunk and the branches are proportional to the number of mutations. (**C**) Mutational signatures landscape of eight melanoma samples highlighting the most representative signatures (others = signature contribution < 5%). (**D**) Heatmap illustrating the copy number variation profile of all samples. PST = primary site tumor population, MT = metastatic tumor population.

**Figure 3 ijms-26-01758-f003:**
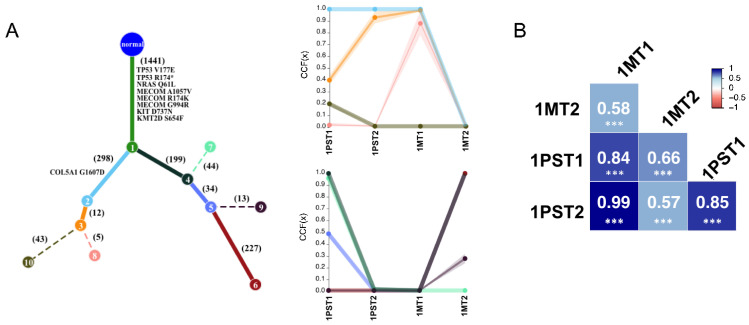
(**A**) Phylogenetic tree built on SNVs of Patient 1 shows the best solution for the evolutionary relationship between clones with different clusters of mutations where each node (numbered) is a cluster of mutations, and each cluster is represented by a different color. Numbers on each branch show the number of mutations distinguishing a cluster. Potential driver genes mutated in the distinction between a clone and the previous are indicated close to the corresponding branch. Solid branches show clusters of mutations that are clonal in at least one sample. Trace plots showing the cancer cell fraction (CCF) for each mutational cluster in each patient sample specific for each of the two main trajectories shown in the phylogenetic tree. Ribbons show the 95% confidence interval, and the centers of bands show the mean cluster CCF estimate. (**B**) Heatmap showing the Pearson’s correlation coefficient between each tumor biopsy. The asterisks indicate a *p* value lower than 0.0001. PST = primary site tumor, MT = metastasis tumor.

**Table 1 ijms-26-01758-t001:** Clinical and pathological characteristics. The order of biopsies follows the clinical history.

Patient	Age at the Time of Biopsy	Sex	Drug Treatment	Biopsies (Site)	Pathology	Biopsy ID
1	63	Female	No drug treatment	Left thigh	Primary nodular melanoma pT4a	na *
	64		low dose Interferon alpha 2b	Lymph node left groin	Lymph node metastasis (1 out of 11 positive)	1MT2
	64		low dose Interferon alpha 2b	Subcutis primary site, left thigh	Subcutaneous local recurrence	1PST
	65		low dose Interferon alpha 2b	Subcutis left thigh	Subcutaneous in transit metastasis left thigh. Flow cytometry DNA index: 1.4.	1MT1
2	89	Female	na	Lymph node right axilla	Lymph node metastasis (3 out of 17 positive), right axilla. Flow cytometry DNA index: 0.9.	2MT
	89		na	Right axilla	Local subcutaneous recurrence Flow cytometry DNA index: 0.9.	2PST
3	65	Female	No drug treatment	Abdomen	Primary nodular melanoma pT4a	3PST
	65		Neoadjuvant chemotherapy **, Interferon-alpha and Interleukin 2	Breast	Subcutaneous in transit metastasis left breast.	3MT

* Tumor biopsy not available at University Hospital of Basel. ** Dacarbazin, Cisplatin, Vinblastine.

**Table 2 ijms-26-01758-t002:** Characteristics of sorted tumor populations.

Patient ID	Biopsy	Population ID	Ploidy (FACS)	Purity Before Sorting (FACS)	Purity After Sorting (WES)
1	Primary site	1PST1	1.7	19%	88%
		1PST2	2.7	41%	95%
	Skin metastasis	1MT1	2.6	80%	91%
	Lymph node metastasis	1MT2	2.7	40%	93%
2	Primary site	2PST	1.9	80%	88%
	Lymph node metastasis	2MT	1.9	88%	96%
3	Primary site	3PST	3.0	87%	91%
	Breast metastasis	3MT	3.1	78%	87%

## Data Availability

The data presented in this study are available on request from the corresponding author due to privacy and ethical reasons.

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
