# Peer review of "Tracking the Evolution of Cutaneous Melanoma by Multiparameter Flow Sorting and Genomic Profiling"

_ijms, 2025, doi:10.3390/ijms26041758_

Round 1
Reviewer 1 Report
Comments and Suggestions for Authors
In this paper, the authors used multiparameter flow-sorting and genomic profiling methods to show the heterogeneity of cancer tissue and difference between primary and metastases tumor sites. This study also shows clonal evolution in melanoma. It is an interesting work. A few comments and questions are listed below:
1. for patient 1, it looks like 1MT2 is earlier than 1PST. The results show that 1MT2 is very different from other biopsies in patient 1. Does it indicate this metastasis is from primary tumor on the left thigh, where the biopsy is not available.
2. Figure 3A. "Trace plots showing cancer cell fraction (CCF) for each mutational cluster in each patient sample specific for each metastasis (top = 1MT1; bottom = 1MT2)". If the clusters are specific for 1MT1 in the top right panel, why the brown line is 0 for 1MT1? Same issue for the green line in 1MT2 panel.
3. Figure 3B, the correlation coefficient was calculated from CNV, the correlation coefficient between 1MT1 and 1PST2 is almost 1. However, in figure 2D, 1PST2 is very similar to 1PST1 instead of 1MT1.
4. Different clonal populations might be at different location of the cancer tissue. The authors find two populations in 1PST but not other biopsy. Does 1PST have a larger size?
Author Response
Reviewer 1:
In this paper, the authors used multiparameter flow-sorting and genomic profiling methods to show the heterogeneity of cancer tissue and difference between primary and metastases tumor sites. This study also shows clonal evolution in melanoma. It is an interesting work. A few comments and questions are listed below:
Comment 1: for patient 1, it looks like 1MT2 is earlier than 1PST. The results show that 1MT2 is very different from other biopsies in patient 1. Does it indicate this metastasis is from primary tumor on the left thigh, where the biopsy is not available.
Response 1: We appreciate the reviewer’s insightful observation. Indeed, Table 1 is sorted according to the chronological order of tumor surgical resections. Sample 1MT2 was resected one year after the primary tumor, which was unavailable for analysis due to the surgical resection being performed at a different hospital, as indicated in Table 1. The relapse sample 1PST was resected 10 months after 1MT2, and with this information, we can only hypothesize that 1MT2 originates from the primary tumor rather than from its relapse. This hypothesis is consistent with the phylogenetic tree in Figure 2B (left), where the 1MT2 sample appears before 1PST. Of note, 1MT2 and 1PST share a high number of clonal mutations in cancer driver genes. This confirms that 1MT2 and 1PST share a common ancestor. We have revised the manuscript to clarify this point. In order to make it more understandable to the reader, we have added the following sentence to the legend of Table 1: The order of biopsies follows the clinical history.'
Please see page 5 lines 147 and page 6, paragraph 2.3, lines 182 and 187
Comment 2: Figure 3A. "Trace plots showing cancer cell fraction (CCF) for each mutational cluster in each patient sample specific for each metastasis (top = 1MT1; bottom = 1MT2)". If the clusters are specific for 1MT1 in the top right panel, why the brown line is 0 for 1MT1? Same issue for the green line in 1MT2 panel.
Response 2: We appreciate the reviewer's comment and agree that the figure legend lacks clarity and requires a more detailed explanation. Our goal was to show the two trace plots based on the phylogenetic tree to better illustrate the clusters driving the two tumor trajectories splitting after cluster 1 (green). This approach was intended to emphasize that the metastatic tumor trajectories were primarily influenced by the two main branches of the tree. We have revised the text accordingly:
Please see page 10, paragraph 2.4, lines 257 and 258
Comment 3: Figure 3B, the correlation coefficient was calculated from CNV, the correlation coefficient between 1MT1 and 1PST2 is almost 1. However, in figure 2D, 1PST2 is very similar to 1PST1 instead of 1MT1.
Response 3: We appreciate the reviewer’s observation and agree with the point raised. The apparent discrepancy between Figures 2D and 3B arises from differences in the data and methods used to generate these plots. Figure 2D originally displayed somatic copy number variations (CNVs) based on total copy number variation status (GL_ASCNA), which uses relatively low thresholds for defining copy number loss and gain. In contrast, To provide a higher robust analysis, Figure 3B is based on Pearson’s correlation coefficients derived from log ratios (GL_LRR), rather than GL_ASCNA.
To ensure consistency between the figures and better reflect the correlation patterns shown in Figure 3B, we have updated Figure 2D to display CNV results derived from log ratios (GL_LRR), instead of the previous ASCNA-based classification. This update improves the coherence of the comparison across figures and aligns more closely with the analysis presented in Figure 3B. We changed the methods, and the figure 2D accordingly.
Please see Figure 2 page 8, page 13, paragraph 4.7, lines 452-454
Comment 4: Different clonal populations might be at different location of the cancer tissue. The authors find two populations in 1PST but not other biopsy. Does 1PST have a larger size?
Response 4: We agree that the size of the tumor could in theory, have an impact on the number of populations found. Analysis of the clinical reports revealed that the primary site tumor of Patient 1 was resected completely. The reported size in the clinical report is 1.6x2.0 cm. However, we do not have this information for the other biopsies of this patient. Due to the missing information and to avoid speculations, we prefer not to include this in the manuscript

Reviewer 2 Report
Comments and Suggestions for Authors
I would like to commend the authors for their excellent work in investigating clonal evolution in melanoma using a combination of fluorescence-activated cell sorting (FACS) and whole-exome sequencing (WES). The study is well-designed, and the integration of multiparameter ploidy sorting significantly enhances tumor purity, allowing for a more precise analysis of clonal heterogeneity. The figures and tables are well-prepared and provide strong visual and quantitative support for the findings. The supplementary files are comprehensive and add valuable depth to the study. This manuscript makes a significant contribution to the understanding of intratumoral heterogeneity in melanoma and provides a robust framework for future studies in this area. I recommend acceptance with only minor revisions
Minor Comments:
- The discussion of clonal evolution could benefit from a brief comparison to other methodologies used for tumor heterogeneity studies, such as single-cell sequencing, to highlight the advantages of the approach used in this study.
- Some figure legends could provide slightly more detail, particularly regarding the sorting parameters used in the FACS analysis, for better clarity.
- While the study mentions three patients, additional details on their clinical characteristics, such as prior treatments or disease stage, would strengthen the context of the findings.
- A short paragraph summarizing the key findings from the supplementary data would be helpful for readers who may not immediately access those files.
- While the study acknowledges the need for a larger cohort, a brief mention of potential technical limitations in FACS sorting (e.g., cell viability concerns) could enhance the transparency of the study.
- A minor language revision would further improve readability, particularly in sections where complex methodologies are described.
This is a high-quality study that provides important insights into melanoma evolution. I strongly recommend acceptance after addressing these minor revisions.
Author Response
Reviewer 2:
I would like to commend the authors for their excellent work in investigating clonal evolution in melanoma using a combination of fluorescence-activated cell sorting (FACS) and whole-exome sequencing (WES). The study is well-designed, and the integration of multiparameter ploidy sorting significantly enhances tumor purity, allowing for a more precise analysis of clonal heterogeneity. The figures and tables are well-prepared and provide strong visual and quantitative support for the findings. The supplementary files are comprehensive and add valuable depth to the study. This manuscript makes a significant contribution to the understanding of intratumoral heterogeneity in melanoma and provides a robust framework for future studies in this area. I recommend acceptance with only minor revisions
Minor Comments:
Comment 1: The discussion of clonal evolution could benefit from a brief comparison to other methodologies used for tumor heterogeneity studies, such as single-cell sequencing, to highlight the advantages of the approach used in this study.
Response 1: We thank the reviewer for this suggestion. We agree that scRNA-seq and spatial transcriptomics are crucial technologies for studying tumor heterogeneity at single-cell resolution. We take this opportunity to cite a paper published in Scienceby Tirosh et al. in 2016, titled "Dissecting the multicellular ecosystem of metastatic melanoma by single-cell RNA-seq." This study revealed significant transcriptional diversity among malignant cells, including variations in cell cycle states, spatial context, and drug resistance programs. In addition, we mention the limitations of this technology compared to our approach in the discussion.
Please see page 10, paragraph 3, lines 293 - 301
Comment 2: Some figure legends could provide slightly more detail, particularly regarding the sorting parameters used in the FACS analysis, for better clarity.
Response 2: We have adopted the labels of the x- and y-axes of the FACS plots. In addition, we have added the following information to the legend of Figure 1: Sorting of the biopsies was performed according to anti-SOX10 positivity (y-axis) and DAPI content (x-axis).
Please see Figure 1 page 4
Comment 3: While the study mentions three patients, additional details on their clinical characteristics, such as prior treatments or disease stage, would strengthen the context of the findings.
Response 3: To the best of our knowledge, the patients do not have received any other treatments than those stated in Table 1. However, our information originates only from the clinical reports of the hospital where the study was conducted. We have added to the patient characteristics section in the Methods that all patients had been diagnosed with Stage IV melanoma.
Please see page 3 lines 98 -99
Comment 4: A short paragraph summarizing the key findings from the supplementary data would be helpful for readers who may not immediately access those files.
Response 4: We have added a short paragraph in the Methods section explaining the Supplementary Data (Supplementary Table 1): “A detailed listing of all called somatic variants in the distinct tumor populations, including allelic frequency, genomic loci, amino acid change, mutation type, and potential affiliation to a known cancer gene set, is included in the Supplementary Data (Supplementary Table 1).”
Please see page 13, paragraph 4.6, lines 445 - 448
Comment 5: While the study acknowledges the need for a larger cohort, a brief mention of potential technical limitations in FACS sorting (e.g., cell viability concerns) could enhance the transparency of the study.
Response 5: We thank the reviewer for this great suggestion to improve the transparency on the limitations of the study. We have added this information in the Discussion section. We also want to highlight that cell viability in particular is not of concern, as we have sorted nuclei from frozen tissues and not from fresh tissue (living cells).
Please see page 12, paragraph 3, lines 363 - 369
Comment 6: A minor language revision would further improve readability, particularly in sections where complex methodologies are described.
Response 6: We thank the reviewer for the valuable comment. We hope that the revisions we have made have enhanced the readability and clarity of the manuscript. However, if the reviewer has specific sections that require further improvement, we would be happy to address them.
This is a high-quality study that provides important insights into melanoma evolution. I strongly recommend acceptance after addressing these minor revisions.
